# Value of 3D printing technology combined with indocyanine green fluorescent navigation in complex laparoscopic hepatectomy

**Jian Cheng**[ID], **Zhifei Wang, Jie Liu, Changwei Dou, Weifeng Yao, Chengwu Zhang**\*

General Surgery, Cancer Center, Department of Hepatobiliary & Pancreatic Surgery and Minimally Invasive Surgery, Zhejiang Provincial People's Hospital (Affiliated People's Hospital, Hangzhou Medical College), Hangzhou, Zhejiang, China

\* zcwzry@163.com

**Data Availability Statement:** Data Availability Statement: Our medical ethics committee imposed ethical and legal restrictions on sharing a de-identified data set, because the data contain

## Abstract

### Background

Laparoscopic hepatectomy (LH) has achieved rapid progress over the last decade. However, it is still challenging to apply laparoscopy to lesions located in segments I, VII, VIII, and IVa and the hepatic hilar region due to difficulty operating around complex anatomical structures. In this study, we applied three-dimensional printing (3DP) and indocyanine green (ICG) fluorescence imaging technology to complex laparoscopic hepatectomy (CLH) to explore the effects and value of the modified procedure.

### Materials and methods

From January 2019 to January 2021, 54 patients with complex hepatobiliary diseases underwent LH at our center. Clinical data were collected from these patients and retrospectively analyzed.

### Results

A total of 30 patients underwent CLH using the conventional approach, whereas 24 cases received CLH with 3DP technology and ICG fluorescent navigation. Preoperative data were compared between the two groups. In the 3DP group, we modified the surgical strategy of four patients (4/24, 16.7%) due to real-time intraoperative navigation with 3DP and ICG fluorescent imaging technology. We did not modify the surgical strategy for any patient in the non-3DP group ($P = 0.02$). There were no significant differences between the non-3DP and 3DP groups regarding operating time (297.7±104.1 min vs. 328.8±110.9 min, $P = 0.15$), estimated blood loss (400±263.8 ml vs. 345.8±356.1 ml, $P = 0.52$), rate of conversion to laparotomy (3/30 vs. 2/24, $P = 0.79$), or pathological outcomes including the incidence of microscopical R0 margins (28/30 vs. 24/24, $P = 0.57$). Additionally, there were no significant differences in postoperative complications or recovery conditions between the two groups. No instances of 30- or 90-day mortality were observed.

potentially identifying and sensitive patient information. Upon request, the request for data should be reached by our medical ethics committee (Email: zryllwyh@163.com; Telephone: +86-571-85893677) or sent to the corresponding author (Email: zcwzry@163.com).

**Funding:** This study is supported by National Key R&D Program of China(http://www.htrdc.com/gjszx) under Grant No.2018YFB1107100; Basic Public Welfare Research Project of Zhejiang Province(http://zjnsf.kjt.zj.gov.cn) under Grant No. LGF20H030011 and Zhejiang Medical and Health Science and Technology Plan(http://www.msttp.com/home) under Grant (No.2021KY473; No.2022KY029; No.2022RC096.). The funders had no role in study design, data collection and analysis, decision to publish, or preparation of the manuscript.

**Competing interests:** The authors have declared that no competing interests exist.

## Conclusion

The optimal surgical strategy for CLH can be chosen with the help of 3DP technology and ICG fluorescent navigation. This modified procedure is both safe and effective, but without improvement of intraoperative and short-term outcomes.

## Introduction

With the dramatic development of the laparoscopic technique, laparoscopic hepatectomy (LH) has become more widely adopted for the treatment of hepatic lesions. This method boasts many great advantages, such as enlarged surgical field, minimal invasion, quick recovery, and good prognosis [1]. However, it is still challenging to use laparoscopy to treat hepatolithiasis, type III hilar cholangiocarcinoma (HCA), and lesions located in segments I, VII, VIII, and IVa due to the complexity of anatomical structures, difficult exposure, limited surgical approach, and poor surgical field. It is also difficult to achieve anatomical resection (AR) and guarantee R0 resection in complex laparoscopic hepatectomy (CLH) [2, 3]. Furthermore, the rate of conversion from CLH to laparotomy may be as high as 40% [4]. Aside from the skill of laparoscopic technique, the main reasons for failed CLH include poor preoperative evaluation and limited real-time intraoperative navigation [5]. Over the last decade, three-dimensional (3D) virtual reconstruction technology and indocyanine green (ICG) fluorescence imaging have been applied to preoperative planning and intraoperative navigation with good outcomes [6, 7]. However, virtual preoperative imaging results continue to be incorrectly interpreted, especially in cases involving complex hepatobiliary diseases [8]. This results in inappropriate surgical procedures being chosen, causing severe perioperative complications. Three-dimensional printing (3DP) technology may be a better tool to help choose the optimal surgical strategy and facilitate CLH [9, 10]. However, this technique has rarely been used in combination with LH due to the high cost and lengthy time required for producing a reliable liver model [11]. In this study, we introduce a method to increase the efficiency and reduce the cost of 3DP liver model creation, while still maintaining high quality. In combination with ICG fluorescent navigation, our modified 3DP models were applied to CLH, and their effects and value explored.

## Materials and methods

### Patient cohort and data collection

From January 2019 to January 2021, 54 patients with complex hepatobiliary lesions were enrolled at our center, of which 30 patients underwent CLH using a conventional approach (non-3DP group) and 24 patients underwent CLH with 3DP technology and ICG fluorescent navigation (3DP group). Clinical data were collected, compared, and retrospectively analyzed. Patients were included in this study if they (1) had preoperative data showing lesions located in segments VII, VIII, IVa, or the caudate lobe, or were diagnosed with type III HCA; (2) did not receive chemotherapy before hospitalization; (3) had proper liver reserve function including Child-Pugh class A status, an ICG 15-minute retention rate < 10%, and a future liver remnant ratio > 40%; (4) had no laparoscopic contraindications after preoperative imaging examination and general condition assessment; and (5) had complete clinical and postoperative follow-up data. Patients were excluded from this study if (1) extensive metastasis or impossible radical resection was found during the operation or (2) they couldn't tolerate being under

the laparoscope. This study was conducted in accordance with the Declaration of Helsinki, and the study protocol was approved by the ethics committee of Zhejiang Provincial People's Hospital (2021QT073). Due to the retrospective nature of this study, patient consent for inclusion was waived.

## Preoperative management

Ultrasonography, enhanced computed tomography (CT), and magnetic resonance imaging (MRI) were routinely performed to evaluate aspects of resectability, including tumor size as well as hepatic segment and surrounding blood vessel involvement. Hepatic functional reserve was assessed using the Child-Pugh classification system, ICG 15-minute retention rate (calculated using the DDG-3300K [Pulse Dye Densito-Graph Analyzer, Nihon Kohde, Japan] after ICG [0.5mg/kg] was injected via the peripheral vein), and future liver remnant ratio (calculated by CT scan). CT and MRI data were collected to produce 3DP liver models in the 3DP group.

## 3DP liver model production

Hepatic segmentation and 3D virtual reconstruction were performed using E3D digital medical modeling software V17.06 (Central and Southern E3D Digital Medical and Virtual Reality Research Center, China) based on each patient's CT Digital Imaging and Communications in Medicine (DICOM) data (Fig 1A). Then, the locations of lesions, as well as the surrounding complex vessels and bile ducts, were analyzed and designed using the open-source slicing software Cura 4.4.1 (Ulitmaker, USA). This software generated G-codes that were identified by SLA (Stereo Lithography Appearance) (SL600, Suzhou ZhongRuiZhiChuang 3D Technology Co., LTD., China) to print liver models. The models were made of photosensitive resin material (ZR680, Suzhou ZhongRuiZhiChuang 3D Technology Co., LTD., China) with a bending strength of 66–73 MPa and fracture elongation rate of 10–15%. Photosensitive liquid materials, which were put in a cylinder after previous deaeration, were solidified layer-by-layer using an ultraviolet control system. We only printed the lesions with blood vessels, bile ducts, and their branches (diameter > 2 mm), excluding any spare liver parenchyma. The model was fully

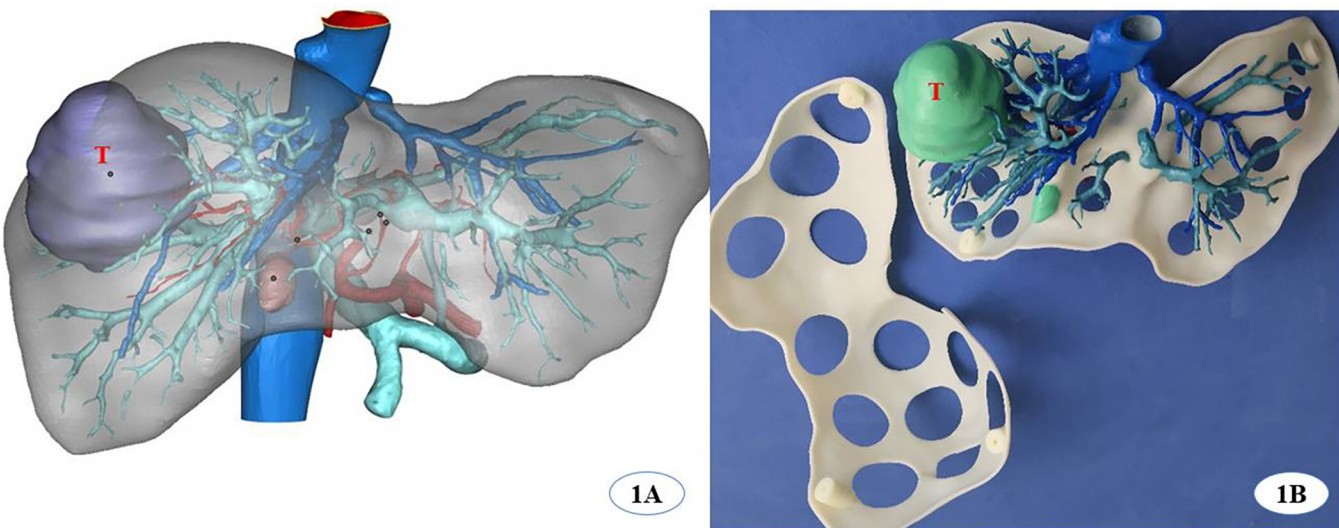

**Fig 1.** 3D virtual reconstruction based on CT data was shown on 1A, A 3D printing liver model modified by hollowed-out designation was shown on 1B. (T indicates tumor, dark blue vessels represent hepatic veins, light blue vessels represent portal veins, white plate full with holes represents liver surface).

hollow with apertures 45–50 mm in diameter. The finalized 3DP liver model was successfully made after curing under a UV mercury lamp and coloring in the post-processing box (Fig 1B).

## Surgical procedure

The surgical procedure varied according to the location of the lesion and its surrounding vessels. For intrahepatic lesions at specific sites, we performed laparoscopic anatomic resection of the hepatic segments; for those classified as type III HCA, we conducted segmented laparoscopic hepatectomy of the extrahepatic bile ducts along with lymph node dissection. The key steps of conventional CLH were performed using the following principles and procedures: (1) the pneumoperitoneum was established with carbon dioxide pressure at 12–15 mmHg, and a total of five trocars were posited according to various hepatectomies; (2) perihepatic ligaments were dissected subsequent to the placement of the first porta blocking band, and intraoperative ultrasound was routinely performed to evaluate potential intrahepatic metastasis and the status of important vessels; (3) after the hepatic pedicle of the segment or lobe to be resected was identified, occluded, and divided, liver parenchyma dissection was performed using an ultrasonic knife and CUSA, where the pringle maneuver was used during hepatectomy if necessary; (4) the segments or lobes were completely mobilized and transected in an en bloc manner; and (5) an abdominal drainage tube was inserted after surgery. In the 3DP group, a patient-specific liver model was brought into the operating room during surgery to help locate and identify important vessels and the range of lesions. ICG (2.5 mg) was injected intravenously after the inflow of the removing segment or lobe was occluded. Intrahepatic ducts were dissected through intraoperative real-time navigation of the 3DP model and ICG fluorescent staining by the fluorescent laparoscope (Figs 2–4). The residual intrahepatic ducts were reconfirmed by considering the 3DP model after operation (Fig 5).

## Postoperative treatment

Postoperative total parenteral nutrition (TPN) was administered before restoring oral intake. Patients received traditional liver-protecting therapy and typically returned to an oral diet 2–3 days after operation. The results of enhanced CT were examined 3–5 days after operation to evaluate intra-abdominal condition. Severe complications were defined as those of Clavien–Dindo grade III or IV [12]. Bile leakage was evaluated and graded according to the definitions of the International Study Group for Liver Surgery (ISGLS) [13].

## Statistical analysis

Statistical analysis was performed using SPSS 23.0 statistical software (SPSS Inc, Chicago, IL). The Chi-squared and Fisher's exact tests were used to evaluate differences between groups for categorical variables, and the Mann–Whitney U test for continuous variables. Survival analyses were performed using the Kaplan–Meier method and the log-rank test. $P<0.05$ was considered statistically significant.

## Results

### Preoperative characteristics

The mean age was 57.9±12.5 years in the non-3DP group and 55.5±12.5 years in the 3DP group ($P = 0.49$). There were 19 male and 11 female patients in the non-3DP group, versus 17 male and seven female patients in the 3DP group ($P = 0.56$). BMI did not significantly differ between the non-3DP and 3DP groups (23.0±3.2 vs. 24.4±4.0, respectively; $P = 0.16$). The counts of patients with hepatocellular carcinoma (HCC), hepatolithiasis, and HCA were 19, 7,

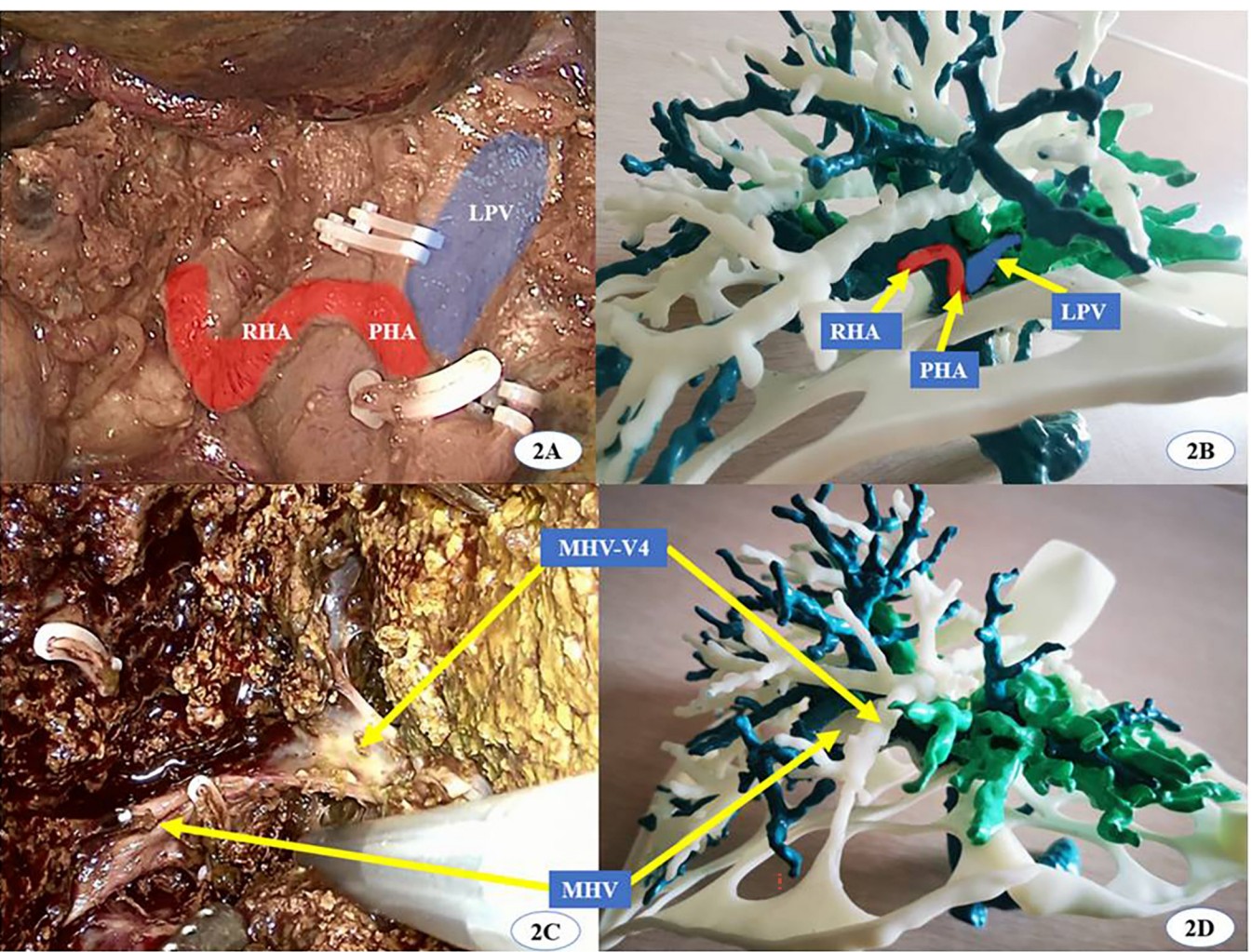

**Fig 2. A modified 3D printing liver model of hilar cholangiocarcinoma was applied in intraoperative navigation.** (RHA: right hepatic artery, PHA: proper hepatic artery, LPV: left portal vein, MHV: middle hepatic vein, MHV-V4: The venous reflux of hepatic segment 4 to MHV).

and 4 in the non-3DP group and 13, 7, and 4 in the 3DP group ($P = 0.69$), respectively. The counts of lesions located in the right anterior lobe, right posterior lobe, middle lobe, right liver, and hepatic hilar region were 7, 7, 3, 9, and 4 in the non-3DP group and 6, 5, 3, 6, and 4 in the 3DP group ($P = 0.80$), respectively. Overall, preoperative data were relatively similar between the two groups (Table 1). The 3DP models were made at a 1:1 size ratio of the actual liver. The mean production time and cost of the models were 56.8 hours and $104.40 USD, respectively.

## Intraoperative data and postoperative outcomes

The 3DP models were consistent with the real intrahepatic structures observed during operation. Preoperative plans in the 3DP group including two right anterior lobectomies, one right posterior lobectomy, and one hilar cholangiocarcinoma radical resection were adjusted to two V segmentectomies, one VII segmentectomy, and one IV segmentectomy, respectively. We modified the surgical strategy for these four patients (4/24, 16.7%) due to real-time navigation using 3DP technology and ICG fluorescent staining in the 3DP group. We did not modify the surgical strategy for any patient in the non-3DP group ($P = 0.02$). The counts of patients

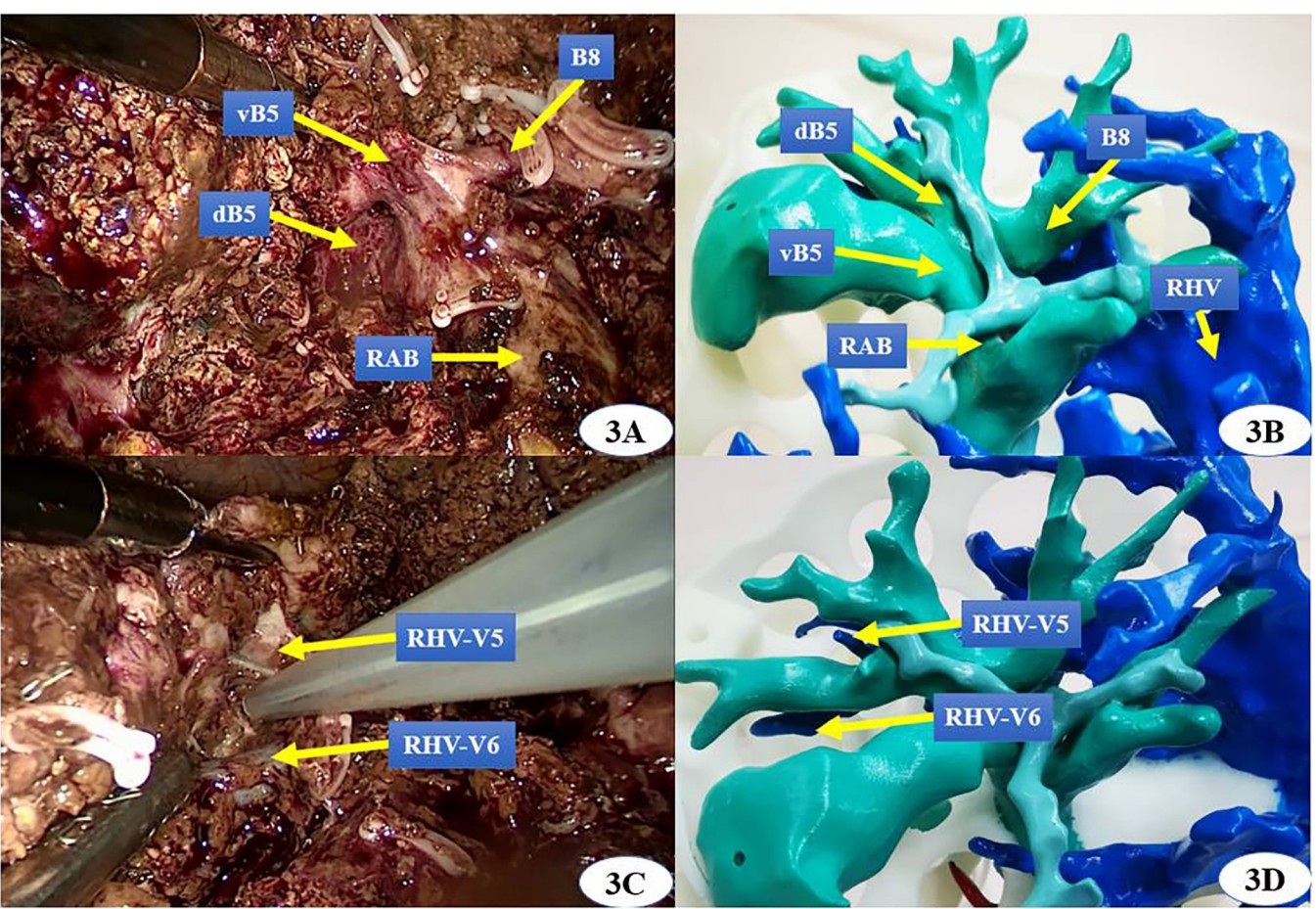

**Fig 3. 3D printing technology was applied in laparoscopic V segmentectomy real-timely.** (vB5: the ventral bile duct of segment 5, dB5: the dorsal bile duct of segment 5, B8: the bile duct of segment 8, RAB: right anterior bile duct, RHV: right hepatic vein, RHV-V5: The venous reflux of hepatic segment 5 to RHV, RHV-V6: The venous reflux of hepatic segment 6 to RHV).

subjected to anatomical segmentectomy, right anterior lobectomy, right posterior lobectomy, mesohepatectomy, right hemihepatectomy, and radical resection of HCA under laparoscopy were 6, 2, 7, 2, 9, and 4 in the non-3DP group and 5, 3, 4, 3, 6, and 3 in the 3DP group ($P>0.05$), respectively. There were no significant differences between the non-3DP and 3DP groups regarding operation time ($297.7\pm104.1$ min vs. $328.8\pm110.9$ min, $P = 0.15$), estimated blood loss ($400\pm263.8$ ml vs. $345.8\pm356.1$ ml, $P = 0.52$), conversion to laparotomy (3/30 vs. 2/24, $P = 0.79$), or pathological outcomes including the incidence of microscopical R0 margins (28/30 vs. 24/24, $P = 0.57$). All HCCs were unifocal. The proportion of patients with HCC who achieved margins $\geq 2$ cm was higher in the 3DP group, but this difference was not significant (61.5% vs. 47.4%; $P = 0.67$). For a total of five patients, surgical strategy was converted to laparotomy due to severe adhesion around important vessels. There were no significant differences in postoperative complications or recovery conditions between the two groups. No cases of 30- or 90- day liver decompensation or mortality were observed in either of the groups. The cost effectiveness did not significantly differ between the non-3DP and 3DP groups ($9431.50 $\pm$9431.50 $\pm$4730.40 USD vs. $10360.70\pm$2899.30 USD, $P = 0.20$). Follow-up was terminated in December 2021, which was when the number of events reached the minimum required for statistical analysis. There was no difference in overall survival (OS) ($P = 0.75$) or recurrence-free survival

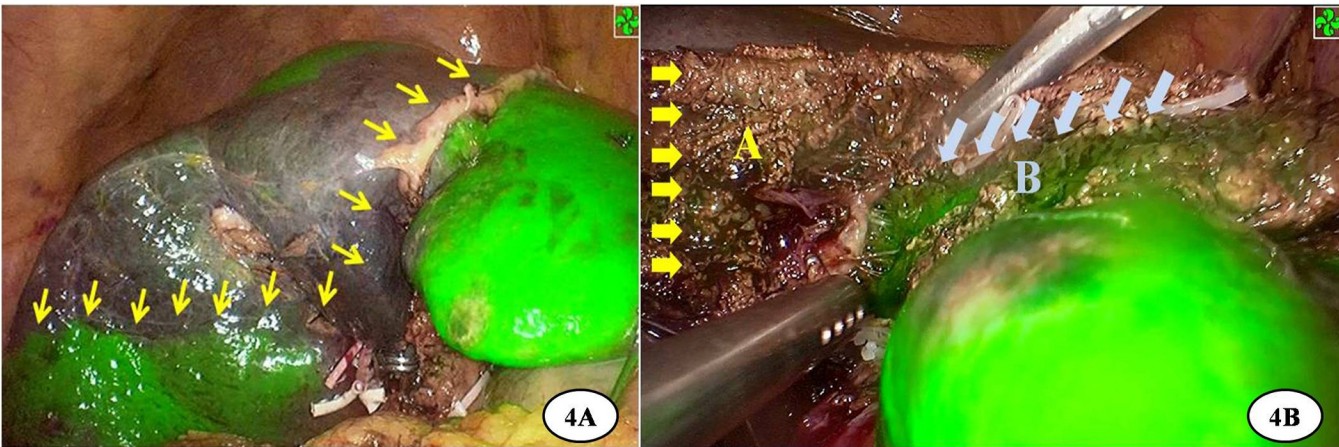

**Fig 4. ICG fluorescent staining was applied to intraoperative navigation.** A: The mark-line (yellow arrows) between the removing liver and the preserving liver. B: Visual contrast in the dissection of liver parenchyma were shown clearly (The green plan B was the preserving side, the other plan A was the removing side).

(RFS) (*P* = 0.25) between the two groups of patients with HCC (Fig 6). Intraoperative data and postoperative outcomes for the two groups are presented in Table 2.

## Discussion

Liver resection is currently accepted as an initial and curative form of treatment for begin and malignant hepatic lesions, even though various alternative treatment choices exist [14]. AR, including systemic removal of the tumor-bearing portal territories, improves local control of the disease by eradicating potential micrometastases via the portal veins [15]. In theory, AR is considered an effective way to treat hepatic lesions. However, the superiority of AR compared to non-anatomical resection (NAR) remains controversial. The most recent studies have shown that AR results in decreased recurrence after the initial hepatectomy. An assessment of recurrence mode and treatment for the recurrence showed that aggressive interventions with curative intent were performed more frequently for intrahepatic recurrence in the NAR group compared to the AR group (42% vs. 10%, *P*<0.001), which led to comparable time-to-

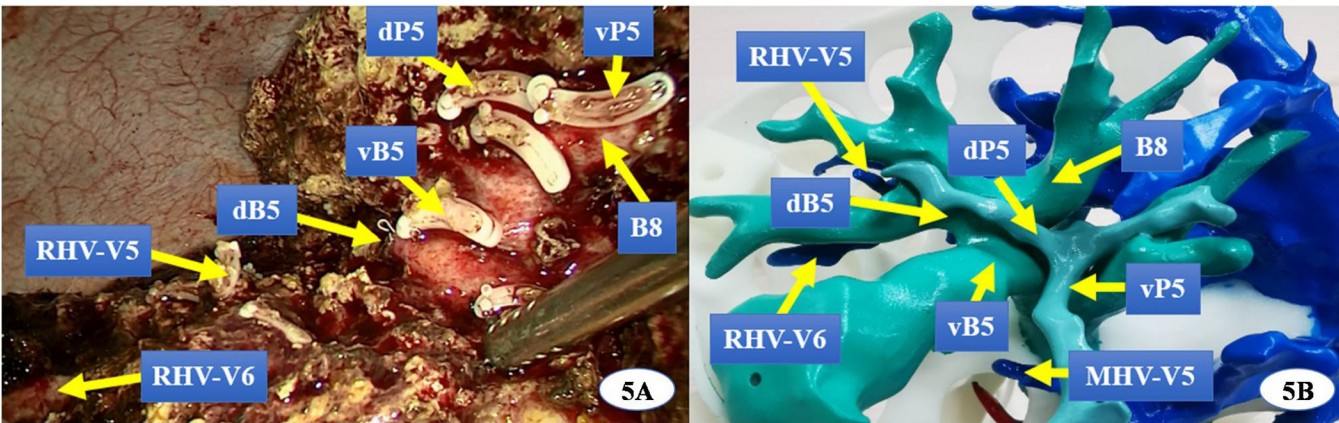

**Fig 5. The actual intrahepatic ducts of surgical field were reconfirmed consistently with the liver model's after operation.** (dP5:the dorsal portal vein of segment 5, vP5:the ventral portal vein of segment 5,vB5: the ventral bile duct of segment 5, dB5: the dorsalbile duct of segment 5, B8: thebile duct of segment 8, RHV: right hepatic vein, RHV-V5: The venous reflux of hepatic segment 5 to RHV,RHV-V6: The venous reflux of hepatic segment 6 to RHV).

**Table 1. Demographics and preoperative characteristics of the non-3DP and 3DP groups.**

| Variables | Non-3DP(n = 30) | 3DP(n = 24) | P value |
|---|---|---|---|
| Background characteristics | | | |
| Age (years) | 57.9±12.5 | 55.5±12.5 | 0.49 |
| Gender (M/F) | 19/11 | 17/7 | 0.56 |
| BMI (kg/m$^2$) | 23.0±3.2 | 24.4±4.0 | 0.16 |
| ASA classification (I/II) | 17/13 | 13/11 | 0.93 |
| HBsAg (+/-) | 21/9 | 12/12 | 0.22 |
| Liver cirrhosis (yes/no) | 9/21 | 8/16 | 0.79 |
| Child-Pugh class (A/B) | 30/0 | 24/0 | |
| ICGR15 (%) | 6.4±1.7 | 5.8±2.2 | 0.14 |
| Liver function test | | | |
| Total bilirubin (umol/L) | 21.3±18.7 | 19.6±11.8 | 0.33 |
| Albumin (g/L) | 37.5±3.7 | 38.8±5.1 | 0.14 |
| ALT (U/L) | 22.4±10.3 | 19.6±9.4 | 0.15 |
| Preoperative diagnosis | | | 0.80 |
| Hepatocellular carcinoma | 19 | 13 | |
| Hepatolithiasis | 7 | 7 | |
| Hilar cholangiocarcinoma | 4 | 4 | |
| Lesion location | | | 0.99 |
| Right anterior lobe | 7 | 6 | |
| Right posterior lobe | 7 | 5 | |
| Middle lobe | 3 | 3 | |
| Right liver | 9 | 6 | |
| Hepatic hilar region | 4 | 4 | |

3DP: 3-dimensional printing; M: male; F: female; BMI: body mass index; ASA: American Society of Anesthesiologists; HBsAg: Hepatitis B surface antigen; ICG R15: indocyanine green retention rate at 15 min; ALT: alanine aminotransferase.

interventional failure and overall survival between the two groups [16]. Thus, the AR surgical technique is more effective than NAR. However, compared to NAR, AR is associated with the adverse features of major liver resection, longer operating time, increased blood loss, and wider surgical margins, and is generally regarded as a more technically demanding operation

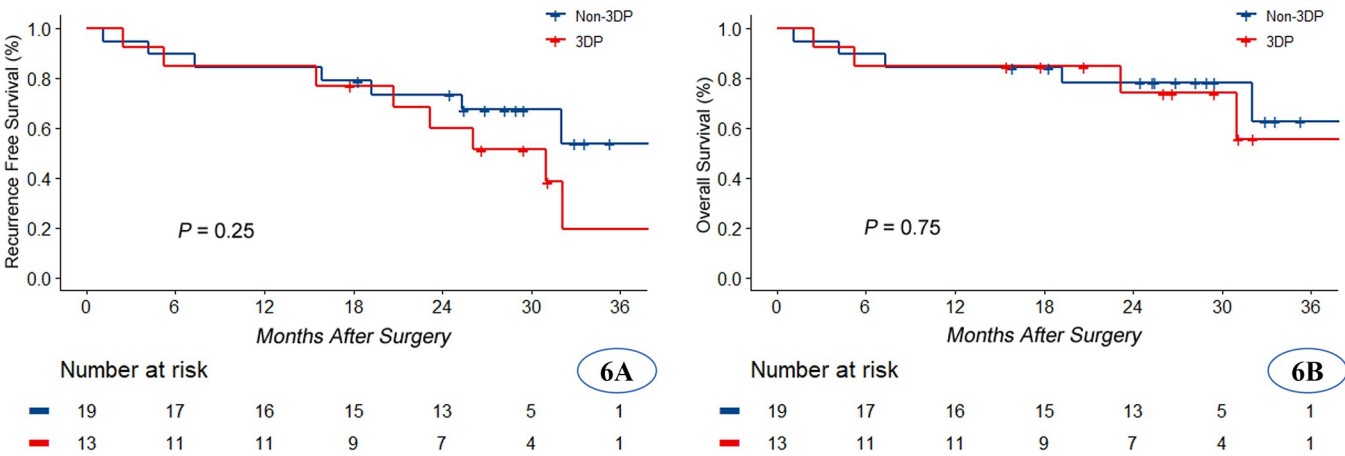

**Fig 6. Recurrence-free survival (RFS) and overall survival (OS) analysis between the 3DP and non-3DP groups of patients with HCC.**

**Table 2. Intraoperative and postoperative characteristics of the non-3DP and 3DP groups.**

| Variables | Non-3DP(n = 30) | 3DP(n = 24) | P value |
|---|---|---|---|
| Surgical strategy modified | 0/30 | 4/24 | **0.02** |
| Surgical approaches | | | |
| Lap. hepatic segmentectomy | 6/30 | 5/24 | 0.94 |
| Lap. right anterior lobectomy | 2/30 | 3/24 | 0.46 |
| Lap. right posterior lobectomy | 7/30 | 4/24 | 0.55 |
| Lap. mesohepatectomy | 2/30 | 3/24 | 0.46 |
| Lap. right hemihepatectomy | 9/30 | 6/24 | 0.68 |
| Lap. radical resection of HCA | 4/30 | 3/24 | 0.93 |
| Operating time (min) | 297.7±104.1 | 328.8±110.9 | 0.15 |
| Blood loss (ml) | 400.0±263.8 | 345.8±356.1 | 0.52 |
| R0 resection | 28/30 | 24/24 | 0.57 |
| Conversion to laparotomy | 3/30 | 2/24 | 0.79 |
| Liver function test of POD 1 | | | |
| Total bilirubin (umol/L) | 30.7±16.3 | 33.4±19.5 | 0.29 |
| Albumin (g/L) | 29.4±4.6 | 30.1±4.4 | 0.28 |
| ALT (U/L) | 243.4±183.9 | 255.5±133.1 | 0.39 |
| Liver function test of POD 3 | | | |
| Total bilirubin (umol/L) | 31.7±18.5 | 39.1±28.6 | 0.13 |
| Albumin (g/L) | 31.1±4.0 | 31.2±3.1 | 0.46 |
| ALT (U/L) | 164.0±147.4 | 215.8±118.9 | 0.08 |
| Liver function test of POD 5 | | | |
| Total bilirubin (umol/L) | 27.6±21.2 | 31.5±29.2 | 0.29 |
| Albumin (g/L) | 33.3±3.2 | 33.2±3.2 | 0.45 |
| ALT (U/L) | 92.0±95.3 | 138.0±125.5 | 0.07 |
| Postoperative complications | 11/30 | 5/24 | 0.33 |
| Bile leakage | 5/30 | 2/24 | 0.62 |
| Intra-abdominal abscess | 3/30 | 1/24 | 0.79 |
| Pulmonary infection | 3/30 | 2/24 | 0.77 |
| Intra-abdominal bleeding | 0 | 0 | |
| Liver failure | 0 | 0 | |
| Postoperative length of stay | 12±5.4 | 8.9±8.7 | 0.53 |
| Hospitalization costs (dollars) | 9431.5±4730.4 | 10360.7±2899.3 | 0.20 |
| Tumor factors of HCC | | | |
| Size (cm) | 4.8±3.4 | 6.1±2.7 | 0.70 |
| Surgical margin (<2cm/≥2cm) | 10/9 | 5/8 | 0.67 |
| AJCC stage (IB/II/IIIB) | 4/9/6 | 4/5/4 | 0.89 |

3DP: three-dimensional printing; Lap: laparoscopic; POD: postoperative day; ALT: alanine aminotransferase; AJCC: American Joint Committee on Cancer.

[17]. Laparoscopic AR is even more challenging, especially for lesions located close to the main hepatic vein, portal vein, or inferior vena cava [18, 19]. In this study, we applied 3DP models and ICG fluorescence imaging technology to LH for the treatment of complex liver lesions, and facilitated the procedures both safely and effectively.

The liver is a solid and opaque organ separated into eight regions known as Couinaud segments with a dense and intricate distribution of blood vessels and bile ducts. LH has become more widely used for the treatment of hepatic lesions due to recent advances in related

instruments and surgical techniques. However, it is still difficult to achieve AR and guarantee R0 resection while using laparoscopic techniques to treat complex hepatobiliary diseases. Aside from the skill of laparoscopic technique, the main reasons for failed CLH include poor preoperative evaluation and limited real-time intraoperative navigation [5]. 3D virtual reconstructive technology based on CT or MRI data has been widely applied to the preoperative planning and intraoperative navigation of CLH with decent short-term outcomes [20]. However, this procedure still has the following undesirable characteristics. First, virtual 3D models lack the aspect of physical touch and space. The sense of touch offers quick visual information transfer by handling a physical object, which is missing when the same images (either 2D or 3D) are displayed on a screen [21]. Second, inexperienced hepatobiliary surgeons may have difficultly comprehending spatial structures in the virtual image, which could result in an improper understanding of the disease state and inappropriate choice of surgical strategy. Third, virtual images are displayed on a 2D plane and overlap with each other. This may result in inaccurate preoperative evaluation and lead to severe perioperative complications, especially for complex hepatobiliary diseases [22]. On the contrary, 3D printing is a technology that transforms a virtual 3D model into a real object, thus helping to solve the problems of virtual 3D technology. 3D printing can produce an anatomical physical model based on the unique characteristics of an individual patient, allowing for comprehensive multi-angle observation of intra-hepatic structures [23, 24]. In our study, we adjusted and oriented the liver models during surgery to match the actual anatomical positions we observed. This helped us to precisely separate important blood vessels and bile ducts (Figs 2 and 3), form an appropriate surgical strategy, avoid accidental damage to intrahepatic vessels, and facilitate CLH both safely and efficiently through real-time navigation. In this study, preoperative plans in the 3DP group including two right anterior lobectomies, one right posterior lobectomy, and one hilar cholangiocarcinoma radical resection were adjusted to two V segmentectomies, one VII segmentectomy, and one IV segmentectomy, respectively. We modified the surgical strategy for these four patients (4/24, 16.7%) due to real-time navigation of 3DP technology and ICG fluorescent imaging. We did not modify the surgical strategy for any patient in the non-3DP group ($P$ = 0.02). We were able to choose the optimal surgical procedures with increased precision and a lower volume of resection through preoperative planning and intraoperative navigation based on the 3DP models, as compared to using 3D simulation reconstruction. All final pathological microscopic R0 margins were negative in 3DP group, while two were positive in the non-3DP group.

3D printing has been widely applied to plastic and reconstructive surgeries with good clinical effects [25–27]. However, the effects and value of applying 3D printing to LH have rarely been reported on and are relatively uncertain, especially due to the lengthy time and high cost required to produce liver models [28, 29]. Currently, it would take about 72–160 hours and $434.50-$869.10 USD to create a simple liver model [30, 31]. In one report, a fully transparent model with three to four levels of intrahepatic duct branches took about 10 days to complete and cost $1552 USD [8]. This sort of model is too time-consuming for preoperative preparation and barely affordable for most patients in China. To account for this, we modified the process of 3DP liver model production, with focus on reducing time and cost while ensuring high quality. Our models were made of inexpensive photosensitive resin materials and printed at a 1:1 size ratio to real liver structures using a hollowed-out design [29]. Only the hepatic lesions with blood vessels, bile ducts, and their branches (diameter > 2 mm) were printed, excluding any spare liver parenchyma. The liver surface was designed with 45–50 mm diameter apertures. We created a modified model that included visualized, tangible, and multi-perspective intra- and extra-hepatic structures, while also reducing both the time and cost of production. From data extraction to final polishing, the model took about 56.8 hours to create and only

cost about $104.40 USD, which is much cheaper than the previously reported models. In Zein's study, the dimension error between the 3DP liver model and the actual liver was less than 4.0 mm, and the intrahepatic blood vessel diameter error was less than 1.3 mm [32]. In our study, the intrahepatic ducts of the 3DP liver models were validated according to their corresponding real livers during operation (Figs 2, 3 and 5). Furthermore, the hospitalization costs for the non-3DP and 3DP groups did not significantly differ ($9431.50±$4730.40 USD vs $10360.70±$2899.30 USD, $P$ = 0.20). These advantages increase the possibility of applying 3DP technology and ICG fluorescence imaging to CLH.

Aside from the skill of laparoscopic technique, successful LH depends on the correct determination of the anatomic boundary and the plane of disconnected liver parenchyma. Large hepatectomies, such as sectionectomies or hemihepatectomies, pose the risk of impairing liver function and may even result in postoperative liver failure, whereas insufficient hepatectomies may spare a greater amount of "at-risk" residual liver parenchyma in which future ischemia and recurrences may occur. In addition, inaccurate intraoperative assessment has commonly resulted in tumor exposure at the surgical margin. The choice of surgical procedure must also consider radicality and the hepatic functional reserve. Although 3DP offers many great advantages, it is still difficult to precisely identify the plane between the removing and preserving hepatic segments using this technology. At present, ICG fluorescent imaging has gained popularity for intra-operative navigation during CLH due to its characteristics of retention and aggregation in hepatic cancer tissues, as well as excretion through the biliary ducts [33]. In our study, after the ligation of the hepatic pedicle in the removing segment, 2.5 mg of ICG was injected through the peripheral vein. Then, the visual contrast liver parenchyma between the removing and preserving hepatic segments was clearly visualized, allowing us to precisely perform hepatic parenchyma disconnection through the real-time navigation of ICG reverse staining (Fig 4). All CLH procedures were successfully performed except for two cases in the 3DP group (2/24, 8.3%) and three cases in the non-3DP group (3/30, 10%), which all involved conversion to laparotomy due to tight adhesion around important vessels as opposed to technical reasons. There were no significant differences between the non-3DP and 3DP groups regarding operating time (297.7±104.1 min vs. 328.8±110.9 min, $P$ = 0.15), estimated blood loss (400±263.8 ml vs. 345.8±356.1 ml, $P$ = 0.52), conversion to laparotomy (3/30 vs. 2/24, $P$ = 0.79), or pathological outcomes including the incidence of microscopic R0 margins (28/30 vs. 24/24, $P$ = 0.57). There were also no significant differences in postoperative complications or recovery conditions between the two groups. No instances of 30- or 90-day mortality were observed in either of the groups. Intraoperative data and postoperative outcomes for the two groups did not significantly differ. The possible reasons for these findings are as follows: (1) the small sample size of this study was too underpowered to show statistical differences between the two groups; (2) CLH is a very complicated and time-consuming operation in itself [34], even with the help of 3DP technology and ICG fluorescent navigation; (3) all surgeries were performed by experienced and highly-skilled laparoscopic surgeons, who could conduct CLH both safely and effectively using traditional methods even without 3DP technology. Overall, we conclude that the use of 3D printing and ICG fluorescence imaging technology in combination with CLH improves resection precision and the preservation of healthy liver parenchyma.

## Limitations

Our study had several limitations worth noting. First, this was a retrospective study with a small sample size. Second, CLH procedures were performed by several experienced surgeons, which may have had an impact on perioperative outcomes (especially operating time, blood

loss, and complications). Third, this study focused on surgical technique innovation and short-term effects rather than long-term oncological outcomes, which is another crucial factor that deserves further investigation. Finally, well-designed randomized controlled trials with larger sample sizes that involve multiple centers should be conducted to verify the advantages of 3DP technology and ICG fluorescence imaging in CLH.

## Conclusions

Our modified 3DP liver model required less time and money to produce compared to previous models, which increases the possibility of its application. With the help of 3DP technology and ICG fluorescent navigation, we may form the optimal strategy for CLH. However, this method didn't improve intraoperative and short-term outcomes. Future studies should focus on long-term oncological outcomes and involve larger sample sizes.

## Author Contributions

**Conceptualization:** Jian Cheng, Zhifei Wang, Jie Liu, Changwei Dou, Weifeng Yao, Chengwu Zhang.

**Data curation:** Jian Cheng, Jie Liu, Changwei Dou, Weifeng Yao.

**Methodology:** Jian Cheng, Jie Liu, Changwei Dou, Weifeng Yao, Chengwu Zhang.

**Resources:** Jian Cheng, Chengwu Zhang.

**Supervision:** Zhifei Wang, Jie Liu, Chengwu Zhang.

**Validation:** Zhifei Wang, Jie Liu, Changwei Dou, Weifeng Yao, Chengwu Zhang.

**Writing – original draft:** Jian Cheng.

**Writing – review & editing:** Jian Cheng.

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
