## [Decision Letter · Decision Letter 0]

23 Feb 2022

PONE-D-21-33418Value of 3D printing technology combined with indocyanine green fluorescent navigation in laparoscopic complex hepatectomyPLOS ONE

Dear Dr. Chengwu Zhang,

Thank you for submitting your manuscript to PLOS ONE. After careful consideration, we feel that it has merit but does not fully meet PLOS ONE’s publication criteria as it currently stands. Therefore, we invite you to submit a revised version of the manuscript that addresses the points raised during the review process.

We look forward to receiving your revised manuscript.

Kind regards,

Gianfranco D. Alpini

Academic Editor

PLOS ONE

Journal Requirements:

Whilst you may use any professional scientific editing service of your choice, PLOS has partnered with both American Journal Experts (AJE) and Editage to provide discounted services to PLOS authors. Both organizations have experience helping authors meet PLOS guidelines and can provide language editing, translation, manuscript formatting, and figure formatting to ensure your manuscript meets our submission guidelines. To take advantage of our partnership with AJE, visit the AJE website (http://aje.com/go/plos) for a 15% discount off AJE services. To take advantage of our partnership with Editage, visit the Editage website (www.editage.com) and enter referral code PLOSEDIT for a 15% discount off Editage services.  If the PLOS editorial team finds any language issues in text that either AJE or Editage has edited, the service provider will re-edit the text for free.

4. Thank you for stating the following in the Acknowledgments/ Funding Section of your manuscript: 

This study is supported by National Key R&D Program of China(2018YFB1107100); Basic Public Welfare Research Project of Zhejiang Province (LGF20H030011); 

Reviewers' comments:

Reviewer's Responses to Questions

**Comments to the Author**

1. Is the manuscript technically sound, and do the data support the conclusions?

Reviewer #1: Yes

Reviewer #2: Yes

2. Has the statistical analysis been performed appropriately and rigorously? 

Reviewer #1: I Don't Know

Reviewer #2: Yes

3. Have the authors made all data underlying the findings in their manuscript fully available?

Reviewer #1: Yes

Reviewer #2: Yes

4. Is the manuscript presented in an intelligible fashion and written in standard English?

Reviewer #1: No

Reviewer #2: No

5. Review Comments to the Author

Reviewer #1: The authors present an innovative approach to surgical resection of difficult liver lesions. The employment of 3D modeling of the vascular network as well as implementation of dye helps surgeons clearly identify the liver tissue for retention and for removal. The following edits should be made prior to publication in PLoS ONE journal:

1. Thorough editing of the manuscript language. Various grammar errors were found in this manuscript and should be reviewed. Notably, words should not start with conjunction words like " But".

2. The discussion should be rewritten (first couple of paragraphs) due to its similarity to the introduction. The discussion is where the authors are able to criticize their work and see how it supports/contradicts other literature. It is this reviewers suggestion that the authors revamp this part of their work.

3. Inclusion of the cost effectiveness of 3D printing + Dye insertion should be included in this manuscript. What are the potential healthcare costs of these difficult surgeries? How much does inclusion of 3D visualization and dye injection help reduce this cost both for clinicians and patients?

Reviewer #2: In this paper 3D printing(3DP) technology and indocyanine green(ICG) fluorescent navigation were applied in laparoscopic complex hepatectomy(LCH) and explored the effects and value. 30 patients underwent LCH by conventional approach and 24 cases received LCH using 3DP technology and ICG fluorescent navigation.

Major comments:

1) data on the presence of liver cirrhosis in liver parenchyma should be provided

2) occurrence of liver decompensation at 30-day and 90- day post-intervention should be provided

3) authors should carefully discuss how much £D printing(3DP) technology and indocyanine green(ICG) fluorescent navigation may imply in term of health costs.

4) value of liver tests may vary among the groups in post surgical period. In particular cholestasis indexes. Please provide these data.

5) English should be revised.

6. PLOS authors have the option to publish the peer review history of their article (what does this mean?). If published, this will include your full peer review and any attached files.

Reviewer #1: No

Reviewer #2: No

---

## [Author Response · Author response to Decision Letter 0]

14 Mar 2022

Dear editor,

Thank you very much for your letter and advice. We have revised the manuscript, and would like to re-submit it for your consideration. We have addressed the comments raised by the reviewers, and the amendments are highlighted in the revised manuscript. Point by point responses to the reviewers’ comments are listed below this letter. This manuscript has been edited and proofread in standard English. We hope that the revised version of the manuscript is now acceptable for publication in your journal.

I look forward to hearing from you soon.

With best wishes, 

Yours sincerely,

Chengwu Zhang

Corresponding author

We would like to express our sincere thanks to the reviewers for the constructive and positive comments.

Replies to Reviewer #1

Specific Comments

1. Thorough editing of the manuscript language. Various grammar errors were found in this manuscript and should be reviewed. Notably, words should not start with conjunction words like " But".

Answer: The manuscript has been edited in an intelligible fashion and written in standard English.

2. The discussion should be rewritten (first couple of paragraphs) due to its similarity to the introduction. The discussion is where the authors are able to criticize their work and see how it supports/contradicts other literature. It is this reviewers suggestion that the authors revamp this part of their work.

Answer: The discussion has been rewritten in the first couple of paragraphs in the revised version to address this issue.

3. Inclusion of the cost effectiveness of 3D printing + Dye insertion should be included in this manuscript. What are the potential healthcare costs of these difficult surgeries? How much does inclusion of 3D visualization and dye injection help reduce this cost both for clinicians and patients?

Answer: Yes, the relevant data were added to state the cost effectiveness of 3D printing and ICG imaging in laparoscopic hepatectomy in the Table 2. in the revised version. 

Replies to Reviewer #2

Specific Comments

1. Data on the presence of liver cirrhosis in liver parenchyma should be provided.

2. Occurrence of liver decompensation at 30-day and 90- day post-intervention should be provided.

Answer: The relevant data were added in the Table 1 and 2 in the revised manuscript.

3. Authors should carefully discuss how much £D printing(3DP) technology and indocyanine green(ICG) fluorescent navigation may imply in term of health costs.

Answer: Discussion about cost effectiveness of 3DP and ICG fluorescent navigation has been added in the revised version.

4. Value of liver tests may vary among the groups in post surgical period. In particular cholestasis indexes. Please provide these data.

Answer: The relevant data were provided in the Table 2 in the revised manuscript.

5. English should be revised.

Answer: The manuscript has been edited in an intelligible fashion and written in standard English.

---

## [Decision Letter · Decision Letter 1]

25 Apr 2022

PONE-D-21-33418R1Value of 3D printing technology combined with indocyanine green fluorescent navigation in laparoscopic complex hepatectomyPLOS ONE

Dear Dr. Zhang,

Thank you for submitting your manuscript to PLOS ONE. After careful consideration, we feel that it has merit but does not fully meet PLOS ONE’s publication criteria as it currently stands. Therefore, we invite you to submit a revised version of the manuscript that addresses the points raised during the review process.

Please address the critique of the additional reviewer.

We look forward to receiving your revised manuscript.

Kind regards,

Gregory Tiao, M.D.

Academic Editor

PLOS ONE

Journal Requirements:

Reviewers' comments:

Reviewer's Responses to Questions

**Comments to the Author**

1. If the authors have adequately addressed your comments raised in a previous round of review and you feel that this manuscript is now acceptable for publication, you may indicate that here to bypass the “Comments to the Author” section, enter your conflict of interest statement in the “Confidential to Editor” section, and submit your "Accept" recommendation.

Reviewer #2: All comments have been addressed

Reviewer #3: (No Response)

2. Is the manuscript technically sound, and do the data support the conclusions?

Reviewer #2: (No Response)

Reviewer #3: Yes

3. Has the statistical analysis been performed appropriately and rigorously? 

Reviewer #2: (No Response)

Reviewer #3: Yes

4. Have the authors made all data underlying the findings in their manuscript fully available?

Reviewer #2: (No Response)

Reviewer #3: Yes

5. Is the manuscript presented in an intelligible fashion and written in standard English?

Reviewer #2: (No Response)

Reviewer #3: No

6. Review Comments to the Author

Reviewer #2: (No Response)

Reviewer #3: Manuscript PONE-D-21-33418R1 is the first revision of a clinical research study analyzing the outcomes of laparoscopic hepatectomy using three dimensional printing (3DP) and indocyanine green (ICG) compared to standard laparoscopic technique. The authors demonstrated that 3DP and ICG technology is useful and allows safe laparoscopic resection with similar times and short term outcomes.

Criticism:

I did not review the initial manuscript. The current version describes the technology well and includes significant amounts of data.

Further information about the tumor stages, unifocality or multifocality (HCC), and distance to the margins (R0) and why they had 2 cases with positive margins? How were the complications managed? Was a drain sufficient for the bile leak?

Total duration of Pringle maneuver(s) for both cohorts would be interesting to compare and evaluate its impact on LFTs.

The introduction and discussion are thorough however several grammatical and spelling errors need to be corrected and the manuscript would benefit from editorial input from a native English speaking person. The authors mention the limitation of their study from an oncological outcomes standpoint. Do they have any results on at least a 6 months - 2 year follow up, given that the patients were included from 2019?

Two minor details:

1. In table 1 the authors mention p values >0.05 for preoperative diagnosis and lesion location. Why did the authors not include the full number as the p value is greater than 0.05, hence not significant?

2. In patients and methods, the authors state "Written informed consent was obtained from all study participants. Due to the retrospective nature of this study, patient consent for inclusion was waived." These two statements are contradictory and the authors need to clarify what was done, whether patients consented or if the consent process was waived.

7. PLOS authors have the option to publish the peer review history of their article (what does this mean?). If published, this will include your full peer review and any attached files.

Reviewer #2: No

Reviewer #3: No

---

## [Author Response · Author response to Decision Letter 1]

7 May 2022

Dear reviewers,

Thank you very much for your constructive and positive comments. Some of these suggestions are exactly what our article's issues are. We have revised the manuscript point by point as follows. We hope that the revised version of the manuscript is now acceptable for publication.

Replies to Reviewer

Specific Comments

1.Further information about the tumor stages, unifocality or multifocality (HCC), and distance to the margins (R0) and why they had 2 cases with positive margins? How were the complications managed? Was a drain sufficient for the bile leak?

Answer: All the HCCs were single solid tumors, the further information was collected and added in the Table 2. in the revised version. 2 cases with positive margins(<2mm) were found in the Non-3DP group at the final histopathology of surgical specimen, that's because the two HCCs were adjacent tightly to important vessels that must be preserved. They were treated by salvage TACE and oral lenvatinib. The patients suffered from bile leak were only treated by keeping drain and had an uneventful recovery with no more complications eventually. Abdominal drainage tube was usually removed about one month after discharge.

2.Total duration of Pringle maneuver(s) for both cohorts would be interesting to compare and evaluate its impact on LFTs.

Answer: Yes, it was interesting. LFTs on the day 1, 3, 5 after operation actually showed no difference between the two groups. The relevant data listed in the Table 2.

3.The introduction and discussion are thorough however several grammatical and spelling errors need to be corrected and the manuscript would benefit from editorial input from a native English speaking person. The authors mention the limitation of their study from an oncological outcomes standpoint. Do they have any results on at least a 6 months - 2 year follow up, given that the patients were included from 2019?

Answer: The manuscript has been corrected in standard English. Follow-up data were collected and added in the Table 2. in the revised version. There was no difference in overall survival(OS) (P=0.75) and recurrence-free survival (RFS) (P=0.25) between the two groups in patients with HCC.

4.In table 1 the authors mention p values >0.05 for preoperative diagnosis and lesion location. Why did the authors not include the full number as the p value is greater than 0.05, hence not significant?

Answer: Thank you for your meticulous review, the p value was calculated by SPSS and corrected.

5.In patients and methods, the authors state "Written informed consent was obtained from all study participants. Due to the retrospective nature of this study, patient consent for inclusion was waived." These two statements are contradictory and the authors need to clarify what was done, whether patients consented or if the consent process was waived.

Answer: Thank you for your meticulous review again, the mistake should not happen. We deleted the contradictory sentence“Written informed consent was obtained from all study participants.” 

With best wishes, 

Yours sincerely,

Chengwu Zhang

Corresponding author

---

## [Decision Letter · Decision Letter 2]

12 Jul 2022

PONE-D-21-33418R2Value of 3D printing technology combined with indocyanine green fluorescent navigation in laparoscopic complex hepatectomyPLOS ONE

Dear Dr. Zhang,

Thank you for submitting your manuscript to PLOS ONE. After careful consideration, we feel that it has merit but does not fully meet PLOS ONE’s publication criteria as it currently stands. Therefore, we invite you to submit a revised version of the manuscript that addresses the points raised during the review process.

We look forward to receiving your revised manuscript.

Kind regards,

Gregory Tiao, M.D.

Academic Editor

PLOS ONE

Journal Requirements:

Additional Editor Comments:

Please address the grammatical issues raised by reviewer three. Additionally, the authors may consider the observation of reviewer 4 who feels that the manuscript is worthy of acceptance but may get better readership in a surgery focused journal.

Reviewers' comments:

Reviewer's Responses to Questions

**Comments to the Author**

1. If the authors have adequately addressed your comments raised in a previous round of review and you feel that this manuscript is now acceptable for publication, you may indicate that here to bypass the “Comments to the Author” section, enter your conflict of interest statement in the “Confidential to Editor” section, and submit your "Accept" recommendation.

Reviewer #2: All comments have been addressed

Reviewer #3: (No Response)

Reviewer #4: (No Response)

2. Is the manuscript technically sound, and do the data support the conclusions?

Reviewer #2: (No Response)

Reviewer #3: Partly

Reviewer #4: Partly

3. Has the statistical analysis been performed appropriately and rigorously? 

Reviewer #2: (No Response)

Reviewer #3: Yes

Reviewer #4: Yes

4. Have the authors made all data underlying the findings in their manuscript fully available?

Reviewer #2: (No Response)

Reviewer #3: Yes

Reviewer #4: Yes

5. Is the manuscript presented in an intelligible fashion and written in standard English?

Reviewer #2: (No Response)

Reviewer #3: No

Reviewer #4: Yes

6. Review Comments to the Author

Reviewer #2: (No Response)

Reviewer #3: The manuscript PONE-D-21-33418R2 has been significantly improved and the data presentation as well. There are still too many spelling / grammatical issues in the manuscript to accept it for publication. I recommend to have a native English speaking person assist the authors with editing the manuscript.

The data is interesting however there are no significant differences between the two analyzed patient groups to make any significant recommendations.

Reviewer #4: I did not have the opportunity to review the earlier version of this manuscript, but this one reads well and was very interesting. The authors are to be commended for their innovative approach to complex laparoscopic hepatectomy; I think their work will be of broad interest to hepatobiliary surgeons worldwide, despite a lack of statistically significant improvements in surgical outcome in their cohort of patients.

The rationale behind my recommendation lies in that I think this manuscript would be of greater interest to a surgical journal, and would be seen by more surgeons in that type of publication.

7. PLOS authors have the option to publish the peer review history of their article (what does this mean?). If published, this will include your full peer review and any attached files.

Reviewer #2: No

Reviewer #3: No

Reviewer #4: No

---

## [Author Response · Author response to Decision Letter 2]

20 Jul 2022

Dear reviewers,

Thank you very much for your constructive and positive comments. We have revised the manuscript point by point as follows. This manuscript has been edited and proofread in standard English. We hope that the revised version of the manuscript is now acceptable for publication.

Replies to Reviewer 3

Specific Comments from Reviewer #3: The manuscript PONE-D-21-33418R2 has been significantly improved and the data presentation as well. There are still too many spelling / grammatical issues in the manuscript to accept it for publication. I recommend to have a native English speaking person assist the authors with editing the manuscript. The data is interesting however there are no significant differences between the two analyzed patient groups to make any significant recommendation.

Answer: The manuscript has been edited in an intelligible fashion and written in standard English. Yes, this modified procedure didn't improve the intraoperative and short-term outcomes, but the optimal surgical strategy for complex laparoscopic hepatectomy can be chosen with the help of 3DP technology and ICG fluorescent navigation.

Replies to Reviewer 4

Specific Comments from Reviewer #4: I did not have the opportunity to review the earlier version of this manuscript, but this one reads well and was very interesting. The authors are to be commended for their innovative approach to complex laparoscopic hepatectomy; I think their work will be of broad interest to hepatobiliary surgeons worldwide, despite a lack of statistically significant improvements in surgical outcome in their cohort of patients. The rationale behind my recommendation lies in that I think this manuscript would be of greater interest to a surgical journal, and would be seen by more surgeons in that type of publication.

Answer: Thank you for the approval of our work and encouragement. We have been very impressed by the overall quality of the work published by PLOS ONE. It will be my great pleasure and honor to publish our novel work in the PLOS ONE as well. Thank you very much for your kind recommendation.

With best wishes, 

Yours sincerely,

Chengwu Zhang

Corresponding author

---

## [Editor Report · Decision Letter 3]

27 Jul 2022

Value of 3D printing technology combined with indocyanine green fluorescent navigation in complex laparoscopic  hepatectomy

PONE-D-21-33418R3

Dear Dr. Zhang,

We’re pleased to inform you that your manuscript has been judged scientifically suitable for publication and will be formally accepted for publication once it meets all outstanding technical requirements.

Kind regards,

Gregory Tiao, M.D.

Academic Editor

PLOS ONE

Additional Editor Comments (optional):

The authors nicely addressed the minor issues raised in the previous review
---

## [Editor Report · Acceptance letter]

2 Aug 2022

PONE-D-21-33418R3 

Value of 3D printing technology combined with indocyanine green fluorescent navigation in complex laparoscopic hepatectomy 

Dear Dr. Zhang:

I'm pleased to inform you that your manuscript has been deemed suitable for publication in PLOS ONE. Congratulations! Your manuscript is now with our production department. 

Kind regards, 

on behalf of

Dr. Gregory Tiao 

Academic Editor

PLOS ONE